# Climate Change Migration and Displacement: Learning from Past Relocations in the Pacific

**Tammy Tabe** 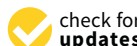

Pacific Centre for Environment and Sustainable Development (PaCE-SD), Laucala Campus,
The University of the South Pacific, Suva, Fiji; tammy.tabe@usp.ac.fj; Tel.: +679-323-2897

**Abstract:** It has been projected that the single greatest impact of environmental changes will be on human migration and displacement. Migration has been extensively discussed and documented as an adaptation strategy in response to environmental changes, and more recently, to climate change. However, forced relocation will lead to the displacement of people, and although much has been written about it, very little has been documented from the Pacific Islands perspective, especially by communities that were forced to relocate as a result of colonialism and those that have been forced to migrate today as a result of climate change impacts. Using the Gilbertese resettlement from the Phoenix Islands to the Solomon Islands, in particular, Wagina Island in the 1960s as a case study of forced relocation and displacement of Pacific Islands people during the colonial period, this paper aims to underline some of the important lessons that can be learned from this historical case to inform the present and future challenges of climate change migration and displacement. Without dismissing migration as a coping strategy, the paper argues that the forced relocation of people from their home islands as a result of climate change will lead to displacement. It accentuates that in the case of Pacific Islands, forced relocation will lead to displacement if they are forced to leave their land because of their deep relationship and attachment to it. The paper also emphasizes the need to acknowledge and honor Pacific Islands' voices and perceptions in discourses on climate change migration and displacement at national, regional and international forums.

**Keywords:** climate change migration; adaptation; displacement; forced relocation; forced migration; Gilbertese people; Phoenix Islands; Wagina Island

## 1. Introduction

Mobility has always been in the nature of Pacific Islanders, and people move for various reasons. During the colonial period, movements of Pacific Islanders have often been associated with economic and political interests of Colonial Empires (Maude 1968; Silverman 1971; Tabucanon and Brian 2011; Tabe 2011; Connell 2012; Edwards 2013; Campbell 2014; McAdam 2014; Teaiwa 2015). Some of these precolonial movements have resulted in the forced relocation and displacement of Pacific Islands people. For instance, in 1945, the population of Banaba was forcefully relocated to Rabi Island in Fiji by the Administration of the Gilbert and Ellice Islands Colony (GEIC) so the British Phosphate Company (BPC) could continue mining the island without resistance from the Banaban community (Hermann 2005; Teaiwa 2015). In 1946, the inhabitants of Bikini atoll were also forcefully relocated to Rongerik, an uninhabited atoll in northern Marshall Islands so the United States of America could use their home island as a nuclear testing ground (Kiste 1977). In the late 1930s, groups of Gilbertese families were relocated from the Southern Gilbert Islands to Phoenix Islands as a result of overpopulation and land hunger. In the mid-1950s and early 1960s, these groups of Gilbertese families were resettled again from the Phoenix Islands to the Solomon Islands as a result of periodic droughts and environmental degradation (Maude 1968; Knudson 1964; Meyen 1992; Tabe 2011).

Other forced relocations that have occurred in the Pacific Islands had been caused by natural hazards such as volcanic eruptions and tsunamis. While at times these movements were temporary, other times, they were permanent. For example, in 1946, the population of Niuafo'ou was forced to relocate as a result of volcanic eruptions on the island of Eua in southern Tonga. Similarly, in 1950, a small population of Ambrym in Vanuatu was forced to relocate to the central island of Efate after a major volcanic eruption on the island. Although, people were displaced during these events, many of them later returned to their home islands to rebuild their lives (Connell 2012). In recent years, tsunamis have also caused both temporary and permanent relocation and displacement of people in the Pacific. For instance, in 2007, parts of the Western Province in Solomon Islands were devastated by a tsunami that struck Gizo town and nearby coastal communities. The tsunami forced many of the coastal communities including the Gilbertese community on Gizo to relocate uphill. However, many of them later returned to the coast to rebuild their homes while others remained on the hilltop and formed new settlements (Zamora et al. 2011; Hagen 2013). Similarly, in 2009, Upolu in Samoa was also devastated by a tsunami that caused extensive damages to the islands infrastructure and forced many of the coastal communities to relocate inland and established new settlements (EERI Special Earthquake Report 2010).

The forced relocation and displacement of people in the Pacific has also been driven by climate change, and it is anticipated to increase in the future. Scientific studies have projected an increase of surface and ocean temperatures, intensity of tropical cyclones, wind speed, rainfall, storm surges, and increase of sea-level rise, coastal inundation, droughts, and other climatic related activities (Mimura et al. 2007; Hulme 2009; Goldring 2015). Many of these impacts have already materialized and have affected Pacific Islands' communities. Tropical cyclones have intensified and their impacts have resulted in extensive damages to national infrastructure and homes, and have also led to the forced relocation and displacement of many people. For instance, in 2015, Category 5 tropical cyclone Pam impacted Vanuatu and caused severe damages to the country's national infrastructure and homes, which was estimated to be equivalent to US$449.4 million. The cyclone also affected many communities and displaced thousands of people in the country (IOM 2015). In 2016, Category 5 tropical cyclone Winston severely affected Fiji and resulted in extensive damages of national infrastructure and homes worth US$0.9 billion. The cyclone uprooted houses and devastated villages across Fiji and displaced many of the people (Government of Fiji 2016).

Heavy rainfall and flash flooding have also affected and displaced many communities. For instance, in 2014, heavy rainfall caused major flash flooding in the Mataniko River, which affected Honiara, the capital town of Solomon Islands. While it caused severe damages to national infrastructure and homes estimated to worth US$109 million, it also led to the forced relocation and displacement of settlements located along the river and at river mouth (Government of Solomon Islands 2014). In 2017, the community of Tukuraki in Fiji was also relocated due to Cyclone Evan in 2012, which resulted in a landslide that buried part of the village, and was again affected by cyclone Winston in 2016 (SPC 2017). Climate change impacts such as sea level rise and coastal erosion have also forced the relocation of several Pacific Islands communities. For instance, in 2014, the community of Vunidogoloa in Fiji was relocated 2 kilometers inland from their original village site as a result of sea level rise, coastal erosion, and storm surges (Charan et al. 2017; Tronquet 2015; McNamara and Combes 2015). Similarly, the clearing and construction of the new town site in 2009 to relocate the Taro Township in Choiseul Province, Solomon Islands, which is being threatened by sea level rise and coastal erosion was suspended due to lack of funding (Yeo 2014).

According to the IPCC report, the rate of global mean seal level rise during the 21st century will likely exceed the rate observed during 1971–2010. This means that the global sea level rise is likely to increase from 0.52 to 0.98 meters between 2081–2100 (Church et al. 2013). This increase will have significant impacts on Pacific Islands who are highly vulnerable to climate change (McAdam 2012; Yamamoto and Miguel 2012). Effects such as coastal erosion, saltwater inundation, and contamination of ground water have already been experienced in many low-lying atolls and coastal communities

and will likely intensify in the future. This will pose future threats to sustainable agriculture and food security for many Pacific Islands because of their high dependency on natural resources for subsistence and income. The increase impacts of climate change will also deteriorate natural ecosystems that they will no longer be able to support the populations of the islands (O'Brien et al. 2012; Mortreux and Barnett 2009). Low lying atolls are particularly prone to coastal erosion and inundation because of their physical limitations in terms of land, water, and food resources. These impacts will likely force people to migrate elsewhere for safety and to seek better access to land, livelihood resources, and income (Gordon-Clark 2012; Legatis 2011; Campbell 2014; Oxfam 2012). Studies have also predicted that low lying atolls in the Pacific will no longer be inhabitable long before they become completely submerged (McAdam 2012; Campbell 2014). Among those greatly affected are low-lying Islands such Kiribati, Tuvalu, and the Marshall Islands, and many coastal communities across the Pacific region whose populations are likely to migrate or forced to relocate in the future as a result of sea level rise.

Migration as an adaptation strategy has always been used for coping with environmental changes. Adaptation in the context of migration is the 'process of adjustment to actual or expected climate effects, or to seek to moderate or avoid harm, and to exploit beneficial opportunities' for vulnerable groups of people (Adger et al. 2014, p. 758). However, Campbell (2014) argues that when dealing with migration, it is important to understand the difference between 'migration as an adaptation strategy' and 'migration as displacement'. Migration as an adaptation strategy has positive implications for migrants and occurs at the individual and household levels. Migration in this case is voluntary based on people's choices to migrate elsewhere for better livelihoods and economic opportunities (Mortreux and Barnett 2009). Migration as an adaptation strategy also involves the relocation of vulnerable people from danger zones to safe areas (McAdam 2015). However, migration as displacement has negative implications. It does not provide people the choices or agency to choose whether they want to migrate or to continue to stay where they are, and occurs mainly at the community level. This is accentuated by Hugo (2012) who emphasizes that communities become displaced when it is no longer possible for them to remain in their islands because the ecosystems and the environment in which they depend on for their livelihoods can no longer sustain them. Migration as displacement can also be viewed as maladaptation—the failure to adapt—which reduces the chances of people to remain in their homes and thus forces them to migrate elsewhere (IPCC 2007).

Much has been written by scholars on 'migration as an adaptation strategy' and 'migration as displacement', but what does this means for vulnerable Pacific Islands that are subject to forced relocation in the future. It is important to reflect on past relocation cases in the Pacific Islands, as they provide relevant lessons for present and future challenges of climate change migration and displacement. Using the Gilbertese resettlement to Solomon Islands, in particular to Wagina Island during the colonial period as a case study, this paper discusses some of the lessons that can be learned from this historical case of forced relocation and displacement of Pacific people. Without dismissing migration as an adaptation strategy, the paper argues that forced relocation will lead to the displacement of Pacific Island communities and people because of their deep relationship and attachment to the land. The paper also emphasizes the need for the acknowledgment privileging of Pacific voices and perceptions on issues of climate change migration and displacement at national, regional, and international forums.

## 2. Methodology and Terminology

### 2.1. Study Sites

This paper is based on an ethnographical study of the Gilbertese resettlement to Solomon Islands that was carried out in the Western Pacific Archives in New Zealand and in the Gilbertese communities on Wagina in Solomon Islands. The Western Pacific Archives is housed at the University of Auckland library and contains most of the colonial records of the British Colonial Administration during its colonization of the Western Pacific. Some of these records have been destroyed to safeguard the

reputation of the British Empire. Wagina Island is located in the southeast of Choiseul Province in Solomon Islands (See Figure 1). The island was uninhabited for many years prior to the Gilbertese resettlement so it became registered as a Crown Land under the British Solomon Islands Protectorate (BSIP). The island comprised of three villages; Kukutin, Arariki, and Nikumaroro, which today are populated by the Gilbertese settlers and their descendants.

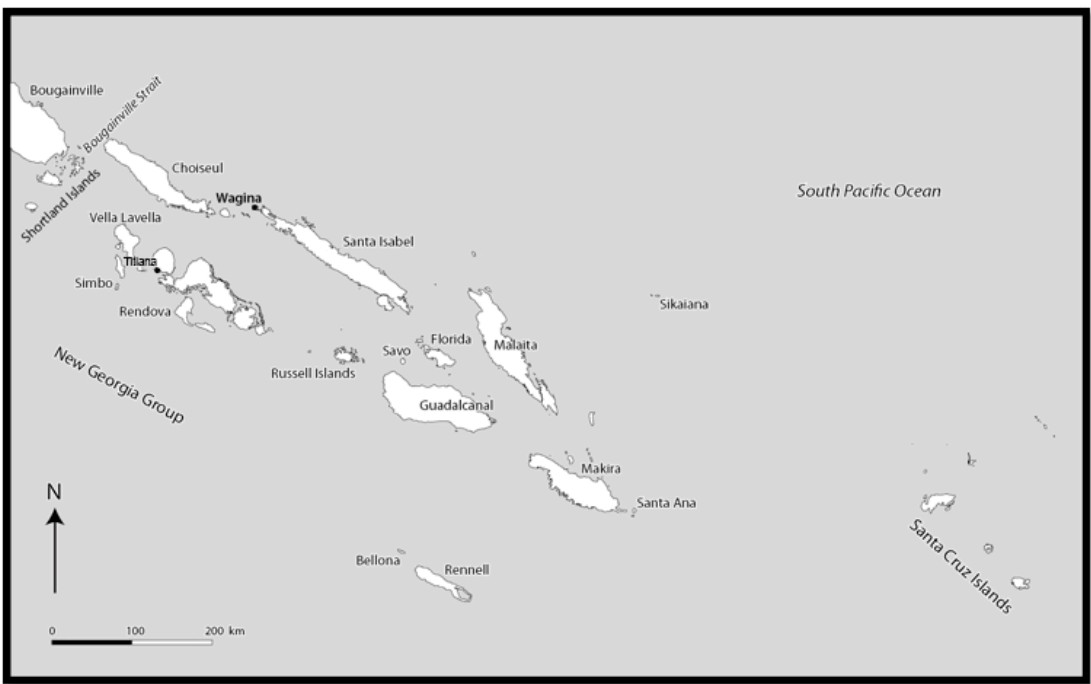

**Figure 1.** Map of Solomon Islands with the location of Wagina Island in the Southeast of Choiseul and Titiana on Gizo Island.

## 2.2. Literature Review

An extensive review of literature on the Gilbertese resettlement from the Southern Gilbert Islands to Phoenix Islands in the late 1930s, and from Phoenix Islands to the Solomon Islands during the mid-1950s and early 1960s was conducted prior to the fieldwork in the Western Pacific Archives and in the Gilbertese communities on Wagina. Limited accounts have been published on the resettlement from the Phoenix Islands to the Solomon Islands, and most of the history that had been written was mainly on the first resettlement of the Gilbertese from the Southern Gilbert Islands to the Phoenix Islands (Maude 1968). Knudson (1964) recorded the accounts of the resettlement of the first group of Gilbertese people to Titiana (Knudson 1964), and Cochrane (1970) recorded the accounts of the resettlement of the second group of Gilbertese to Wagina Island. Much of what was written emphasized 'severe droughts' as the main cause of environmental degradation and the primary driver for the Gilbertese resettlement to the Solomon Islands. The resettlement of the Gilbertese to the Solomon Islands were administered as a form of humanitarian assistance by the GEIC administration to provide opportunities for the Gilbertese people to better livelihoods that will eventually lead to economically self-sufficient communities in the future. It was evident that much of what was published encapsulated colonial perspectives of the Gilbertese resettlements. However, recent accounts of the resettlement to Wagina Island by Meyen (1992) and Tabe (2011) explored for the first time the Gilbertese perceptions of what they believed to have caused their resettlement to the Solomon Islands.

## 2.3. Western Pacific Archives

The use of colonial resources in the Western Pacific Archives (WPA) were strictly regulated so permission to access the Archives was obtained prior to my visit. The Archives were contained in

a Special Collection that regulates the access and use of colonial resources, and I needed consent to access the records. It contained many documents consisting of dialogues, proposals, plans, and reports among colonial administrators about the Gilbert and Ellice Colony (GEIC) and the Gilbertese people. My intention of visiting the archives was to investigate colonial records and uncover information that underline the reasons for the Gilbertese resettlement to Solomon Islands. I had previously learnt from the Gilbertese settlers and from the recently published materials that although periodic droughts were the main reason for the resettlement, the settlers on Wagina Island indicated that the reasons for their resettlement were far too complex than what had been written. They believed that their forced relocation to the Solomon Islands was a result of Britain's nuclear activities on Christmas Island. However, these events were not documented in the history of the Gilbertese resettlement to Solomon Islands or made available in the Archive.

An ethnographical approach that involved close observation and deep analysis of the records, letters, and dialogues in the Archive was employed, not only to uncover reasons for the resettlement but also to understand why the Gilbertese were relocated to the Solomon Islands instead of within the Colony. Many of the records contained information that was fragmented and poorly written so it was difficult to read. The files and letters were not organized in sequence to the date of events and much of the information was missing. A period of two months was spent at the WPA. Photocopying and photography were also used during the archival research to produce copies of the records, which could not be read thoroughly in the Archives.

### 2.4. Ethnographic Fieldwork

Fieldwork was carried out in the three villages on Wagina. The fieldwork was conducted in two phases. The first phase employed the use of focus group discussion and interviews that involved male and female villagers between the age of 50 and 75 years who were anticipated to have a better understanding of the Gilbertese resettlement from Phoenix Islands to the Solomon Islands. The second phase used survey and participant observation to examine the livelihoods of the villagers in terms of subsistence, income, fishing practices, and the land in which they occupy on the island. Results obtained from the first phase of fieldwork will be used in this paper to highlight stories and perceptions of the Gilbertese on Wagina about their resettlement. Prior to the fieldwork, permission was sought from the church ministers in Arariki and Nikumaroro, and from the Catholic priest in Kukutin to conduct research in their communities. The focus group discussions were conducted on Sundays after the church service and mass in each of the villages because it was the only day that most of the villagers were available and free from daily chores. The focus group discussions in each of the three villages were held in the *maneaba*—the Gilbertese meeting house, and involved about 30–50 participants, both women and men. There were also teenagers and children present but they all sat quietly and listened to the conversations. The elderly men were given the privilege to share their stories first before the women contributed to the conversation. The setting of the focus group discussions was open to allow young people and children to also listen to the history of the resettlement.

Interviews were conducted subsequent to the focus group discussions in each of the villages. A total of 20 participants that included both men and women were identified based on their age, gender, and roles in the villages for the interviews. The 20 participants included ten women and ten men, between the age of 50 to 75 years. The primarily purpose of the interviews was to allow the participants to reflect and share their own personal stories, experiences, and perceptions of the resettlement to Wagina, and what they believed to have caused them to resettle to Solomon Islands. The interviews were guided by a set of questions related to the topic of discussion.

### 2.5. Terminology

The following concepts will be used based on their definitions for the purpose of this paper. *Migration* as the process of moving within or across borders, temporarily, seasonally, or permanently. It is commonly associated with an element of choice and is often considered to be voluntary in

nature, or sometimes forced; but in this be used to describe the voluntary movement of people (ODI and UNDP 2017). *Displacement* refers to situations where people are forced from their homes as direct result of slow or quick onset events (ODI and UNDP 2017). In this paper, it will be used to refer to groups of people and communities that are being forced to relocate from their islands as a result of sea level rise, lack of agency and participation in the decision processes, and forced removal from their land. *Resettlement* refers to the process that describes the large-scale movements of people whether voluntarily or forced as a result of development or political projects as replacement of their assets, livelihoods, land and resources to maintain the continuity of their communities and livelihoods. It will be used in this paper to refer to the large-scale movement scheme organized by the GEIC and BSIP administrations. *Relocation* can be voluntary or forced. Voluntary relocation occurs when there is still freedom to choose between realistic options, whereas Forced Relocation takes place when the freedom to choose from the realistic options is no longer available (McAdam and Ferris 2015). *Planned relocation* refers to a solution-oriented measure facilitated by the state in which a community is physically moved resettled in another location. It differs from evacuation even though it may play a role after the evacuation, especially in circumstances where the places of origins are no longer suitable for living (UNHCR 2014).

## 3. The Gilbertese Resettlement: From Gilbert to Phoenix Islands, and to the Solomon Islands

The following history of the Gilbertese resettlement (see Figure 2) has been compiled based on findings from the Western Pacific Archives, literature review, and research on Wagina Island.

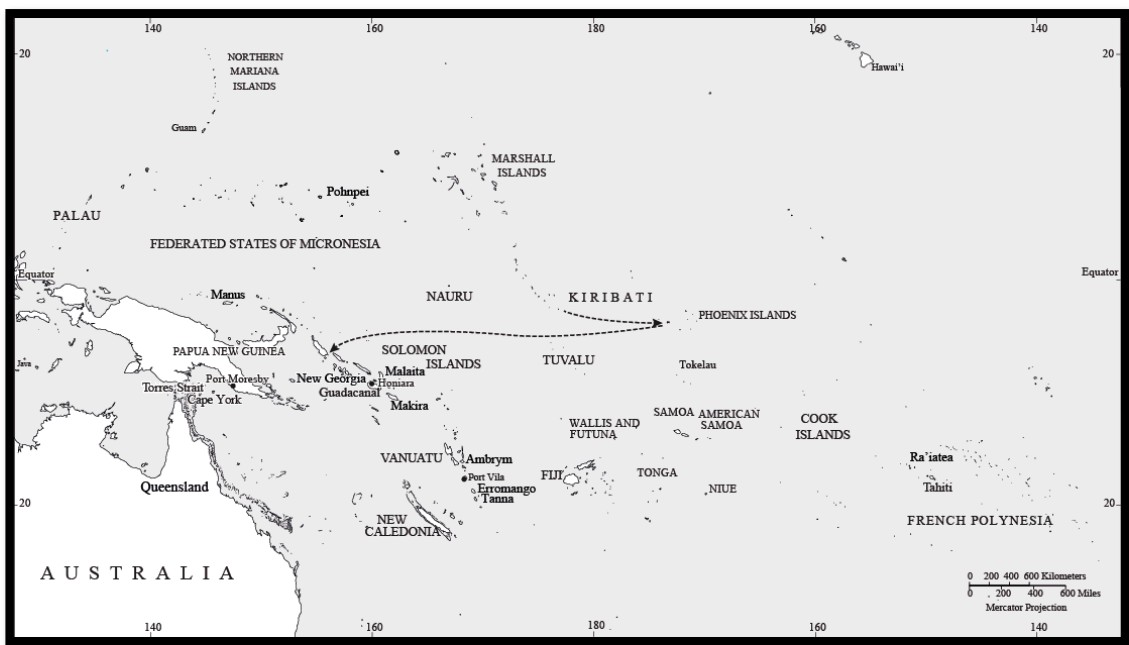

**Figure 2.** Map showing the Gilbertese resettlement routes from Gilbert Islands to Phoenix Islands, and Phoenix Islands to Solomon Islands.

Overpopulation and land hunger became major issues during the initial colonization of the Gilbert Islands. Despite efforts of the GEIC administration to resolve these problems, it was evident that it would only create more problems for the Gilbertese people in the long term. As a result, the administration saw that the only feasible solution to the problems was to redistribute the surplus population to uninhabited islands in the Colony. In October 1937, an administrative tour was conducted throughout the Colony to investigate the suitability of the islands for the resettlement of the surplus Gilbertese population. The Phoenix Islands provided an ideal location for the Gilbertese population because they consisted of a few uninhabited islands that could support large populations, and they also

had similar environmental and climatic conditions with the Gilbert Islands (Maude 1968). In the same year, an expedition team set out to investigate the Phoenix Islands for the possibility of permanent settlement and the number of people each island could support (Knudson 1964). The islands of Hull, Gardner, and Sydney were identified to be suitable for the Gilbertese settlement. One of the important activities carried out during the expedition was the christening of these islands. Hull was named Orona, Gardner was named Nikumaroro, and Sydney Island was named Manra.

The surplus population of the Gilbert Islands was anticipated to be relocated to Phoenix Islands, but priority was given to families from Nonouti, Nikunau, Onotoa, Tamana, Tabiteuea, Beru, and Arorae in the Southern Gilbert that were considered to have high poverty rates, land shortages, be overpopulated, and in need of immediate resettlement. Although, many refused to leave their home islands, they were persuaded of the adequate land and better life in the Phoenix Islands. Between 1939 and 1941, groups of Gilbertese families were relocated from the Southern Gilbert Islands to Phoenix Islands under the Phoenix Islands Resettlement Scheme. The scheme was conducted with the hope to provide opportunities for the Gilbertese families to access adequate land and to have better livelihoods, while reducing the population pressure on limited land and resources in the Gilbert Islands. However, the resettlement scheme was suspended in 1941 due to the outbreak of World War II (Maude 1968). The impacts of the war on many of the islands were severe. Mining infrastructures on Banaba were destroyed, including most of the coconut trees on which people depended for subsistence and income; and it would take some time for the islands to rehabilitate from the damages caused. The phosphate and copra industries were the only backbone of the Colony's economy and their slow recovery after the war was recognized by the GEIC administration to likely result in increased economic expenses, if the islands were to be rehabilitated from the damages caused by the war.

It also became apparent after the war that a poor choice had been made about the Phoenix Islands. The islands turned out to be highly vulnerable to long periods of drought and the lack of food and water resources proved that they were inadequate to support the Gilbertese growing population. Manra at that time was largely affected by droughts and the elders had requested the GEIC administration to relocate them elsewhere. With its dwindling revenue, the GEIC administration became concerned with the management of the Colony, in particular the Phoenix Islands if it were to remain operative. The GEIC administration saw the management of Phoenix Islands would only incur further expenses due to its isolation from Tarawa and increased costs of transportation and relief supplies. Seen as a potential and major economic threat and burden for the GEIC administration in the long-term, the Phoenix Islands Resettlement Scheme was declared a failure. This led the GEIC administration to seek potential resettlement sites in other British colonies and territories, but all its enquiries were unsuccessful. At that time, the Lever's Brothers Company had made a proposal to the BSIP administration for the recruitment of laborers to work in its plantations in the Solomon Islands. This was proposed to the GEIC administration, which saw the opportunity as a solution to its problem. Negotiations were held immediately between the GEIC and the BSIP administrations which were also made easier because both administrations were part of the Western Pacific High Commission. The BSIP administration agreed to take in at least 2000 Gilbertese settlers from the Colony, especially the population of Manra, with prospects of permanent settlement. This eventually led to the facilitation of the large-scale resettlement of the Gilbertese people from Phoenix Islands to Solomon Islands.

The relocation of the entire population of Phoenix Islands was viewed as a solution for both administrations. It would relieve population pressure in the Colony and prevented further economic costs for the GEIC administration from the management of Phoenix Islands. At the same time, the Gilbertese resettlement to the Solomon Islands would provide an additional population and a source of labor for the Lever Brother's copra plantations and contribute to the Protectorate's economy through cash crops. The resettlement to the Solomon Islands was conducted in two waves jointly by the GEIC and the BSIP administrations. The first wave was facilitated under the Titiana Resettlement Scheme and it was conducted through several phases between 1955 and 1958. The population of Manra was relocated to Titiana on Gizo Island in the Western District, today known as the Western Province

([Weber 2016](#); [Tabe 2016](#)). Prior to the resettlement, a pilot team comprised of GEIC administrators and several Gilbertese men travelled to the Protectorate to identify islands that would be suitable for the Gilbertese settlement. Titiana, Wagina, and Tetepare, all three sites located in the Western District at that time were identified as potential locations for the Gilbertese settlements. Of the three sites, Titiana was selected as the most suitable for the Manra population due to its location along the coast and adjacent to a fringing reef, and its close proximity to the Western District headquarters in Gizo town.

The second wave of resettlement was carried out subsequent to the successful resettlement of the entire population of Manra to Titiana. The resettlement was administered under the Wagina Resettlement Scheme whereby the remaining population of the Phoenix Islands on Nikumaroro and Orona were relocated to Wagina Island, which today is part of Choiseul Province. The resettlement was planned in two phases and anticipated to be carried out from 1963–1964 given the emergency nature of the scheme. The scheme was intended to be carried out in different stages to allow the clearing of the village sites and the gradual absorption of the settlers into their new homes. Prior to the resettlement, a meeting was conducted by the Colony officers in the islands of Nikumaroro and Orona, to which the Gilbertese people were informed about the resettlement and the opportunity of a better life in the Solomon Islands. Many of the Gilbertese people did not want to be relocated to the Solomon Islands, especially the elderly people who were not keen on rebuilding their homes and establishing themselves in a new environment. Others were curious to go and see Solomon Islands but had no intentions of settling there. After the meeting, a mandate was issued to the people on both islands that everyone should be relocated to the Solomon Islands and that no person should remain behind. The mandate significantly stressed that those who decided to remain in the islands of Nikumaroro and Orona would be of great economic burden to the GEIC administration, and that it would not be responsible for their welfare; thus, the entire population on both islands was persuaded to be relocated to Wagina ([Tabe 2016](#)).

The Gilbertese people encountered many challenges when they arrived on Wagina given the geographical, environmental, and cultural differences of the islands. Wagina was nothing like the atoll islands in the Phoenix. The island was densely forested and swampy, and characterized by a coastline with rocky limestones that made walking unbearable. Clearing of the village sites was still underway when the entire Gilbertese population on Nikumaroro and Orona arrived on Wagina. The people were accommodated in communal housings and were encouraged to build their houses immediately after the allocation of land so that their families could move out of the housings. Three acres of land within the village area was allocated to each head of households including couples that were newly married, and an additional 10 acres in the mountainous area of the island for agricultural purposes. The Gilbertese were expected to develop their land subsequently to the allocation of plots so that they could sustain their families, but they lacked experience in clearing land with big trees, and had to work hard to develop the land for agricultural purposes.

The isolation of the island from the Western District headquarters in Gizo also raised concerns among the Gilbertese, especially the delay of assistance provided by the District during the settlement of Wagina. At that time, the ocean environment between Wagina and Gizo was unfamiliar to the Gilbertese so they were reluctant to journey in small boats between the two locations. The indigenous people from the nearby islands were also very different in their physically and culturally, and many of the Gilbertese made no efforts in getting to know the people because they were afraid of them. Resources were abundant in fishing areas around the island but the lack of coconut trees was an issue for the Gilbertese who were highly dependent on coconuts for nourishment and income. The prevalent of malaria on the island also affected many of the Gilbertese who had no knowledge of the disease and believed that they were being possessed by spirits of the land. The neighboring indigenous communities believed Wagina was haunted by spirits which had prevented people from being able to settle on the island. The lack of environmental and cultural awareness of Solomon Islands also created an enclave Gilbertese community and did not encourage integration of people into the larger Solomon Islands community. The hard labor, lack of political will and access to economic opportunities, and the

struggle to re-establish themselves in a new environment, and with challenges of not having absolute rights over the land in which they were allocated on Wagina, created a sense of displacement among the Gilbertese people who resented the GEIC administration for bringing them to Solomon Islands—a place unknown to them. However, the Gilbertese on Wagina have continued to live as a community and observed their cultures and languages, and have also gradually integrated into the larger Solomon Islands society through intermarriages, education, employment, and the adaptation of the Solomon Islands way of life (Tabe 2016).

## 4. Lessons from the Gilbertese Resettlement

The Gilbertese resettlement to the Solomon Islands presents both negative and positive implications of forced relocation and displacement. Although it is unclear whether or not the Gilbertese resettlement to Solomon Islands was successful, it nevertheless provides important lessons for future climate change migration and relocation of Pacific Islands, especially those that will be forced to relocate from their home islands as a result of sea level rise. According to the World Bank, the success of a resettlement demonstrates a situation where the settlers' livelihood conditions are at least minimally restored (Voutira and Barbara 2000). However, resettlement success is not only achieved as a result of economic development, but also through genuine establishment of the community and the transfer of a sustainable resettlement process to the later generation of settlers, as new institutions are formed (Scudder 2005). With a prospective view of future relocation of Pacific Islands as a result of climate change, Tabucanon (2012) argues that it is important to re-evaluate how a successful resettlement is defined, without neglecting the economic and social costs involved, while aiming at the long-term sustainability of social and community fabrication.

A number of frameworks and models have also been designed and implemented to assist effective and successful resettlement projects. Cernea's model and conceptualization of resettlement as a process that often involves the risks of being landless, homeless, jobless, marginalized, having a lack of food security, loss of access to common property resources, and increased health issues and social distraction has been used to assist in the evaluation of prospective environmental migration in the Pacific Islands (Cernea 2004). It has been used as a framework to assess certain resettlement features that may prove successful while also taking into account the different cultural and social criteria that may be viewed by settlers as what truly constitutes success significant for possible future resettlements. However, a thorough evaluation of the Gilbertese resettlement to Solomon Islands has yet to be conducted but the movement itself provides important lessons that can be used to inform decisions for Pacific Islands that may be subject to future relocation as a result of sea level rise. There are six primary lessons that can be learned from the Gilbertese resettlement.

### 4.1. Planning and Preparations

It was clear that not a lot of thought was put into the planning and preparation of the Gilbertese resettlement to Solomon Islands, especially in the relocation of an ethnically different group of people from atoll islands to a high island in Solomon Islands. The Gilbertese resettlement was conducted as an emergency measure by the GEIC administration to evacuate the entire population of Phoenix Islands to Solomon Islands with intentions to provide them an opportunity to improve their livelihoods in a new environment. Unlike the Titiana Resettlement Scheme, which involved considerable planning and preparation, the Wagina Resettlement Scheme lacked preparation and proper planning. The resettlement involved a large-scale scheme, which comprised of more than 1000 settlers who were relocated directly from Phoenix Islands to Wagina between 1963 and 1964. This would also cut relocation costs of moving a large number of people between the Colony and Protectorate in several phases over a period of years. However, the island had not been prepared adequately to absorb a large number of Gilbertese who had no familiarity with their new environment. The Gilbertese had to work intensively to re-establish themselves in their new homes. Most of the old men on Wagina indicated that the hard labor they experienced during the clearing of village sites

and land plots on Wagina was nothing like they had experienced before in their lives. One of the old man said '*the trees were too big and we did not know how to cut them down with knives and axes given to us because we have never conducted such heavy labor on the atolls where we came from*'. If resettlements are not planned properly, they will result in the impoverishment of the relocated communities and people (UNHCR 2014). Proper planning and a realistic provision for substantial and properly allocated funding must be prepared in advance. Such funds may be required for land purchases or house constructions and the development of the newly established community (Wrathall 2011).

*4.2. Consent*

Although, the Gilbertese were curious to know more of their new home during the meeting held in Nikumaroro and Orona, they only wanted to see the island, but had no desire of relocating permanently to the Solomon Islands. Limited knowledge of the environment and cultures of the Solomon Islands was also a factor that prevented the elders from wanting to leave their islands and re-establish themselves in a new environment. Many families also did not want to be relocated because they do not want to abandon the graves of their loved ones on the islands. It was apparent that the resettlement to Wagina was decided primarily by the GEIC administration and the settlers had no involvement in the decision. There was lack of consultation with the people about the resettlement and they were not made aware of the reasons for their immediate relocation to Solomon Islands. The meeting was organized primarily to inform the elders in both the islands of the decision to relocate, without consultation or involvement of the people in the decision processes. Although, the settlers refused to be relocated, they were given strict orders that they should all leave and that no person should remain behind. Those who remained behind would not be supported by the GEIC administration; everyone had to be convinced to leave. The settlers were not given the option to decide for themselves but were forced to adhere to the orders and were therefore relocated. It was this lack of informed consent that the Gilbertese felt strongly that their relocation to Wagina was forced on them.

McAdam (2014) writes that consent is important when planning any relocation and it is not the same as consultation and participation. It involves processes of informing, negotiating, and liaising with the prospective settlers concerning the reasons and procedures of the movement while offering alternatives to the relocation that should be considered by authorities. Any relocation should only occur with the consent of the communities and people concerned. The lack of consent usually results in the trauma of displacement and social tensions by the recipient community towards the relocated group (Connell 2012). It is the lack of consent that resulted in sense of displacement, loss of home, land, and identity, and deprivation of opportunities in the new home among the Gilbertese.

*4.3. Challenges Encountered*

The Gilbertese experienced many challenges when they arrived on Wagina because the island which was allocated to them as a new home had not been developed adequately to be able to absorb a large number of people, such that they had to be housed in communal housings, which were overcrowded and lacked privacy for families. The environmental features of the atoll islands contrasted that of Wagina. The island was densely forested without any sandy beaches but was instead characterized by a rocky and limestone coastline. Clearing of the village sites and land plots involved intensive labor for the Gilbertese people. The construction of houses also differed from those built in the atolls because of the difference in weather conditions; therefore, strong materials had to be used to withstand climatic events. The island was also believed to be haunted by spirits; thus, it was uninhabited and many felt that they had been relocated to Wagina purposely to be exterminated. Malaria was one of the major challenges faced by the Gilbertese in Solomon Islands and it caused an unaccounted number of deaths among the Gilbertese during the initial settlement of Wagina, especially coming from non-malarial areas in the Pacific. Many of the settlers were not aware of the symptoms of malaria and assumed on several occasions that they were possessed by the spirits of the land.

The GEIC administration failed to acknowledge that the Gilbertese were not accustomed to densely forested and mountainous islands and that agriculture was not the Gilbertese way of life, and expecting them to contribute towards the Protectorate's economy through cash crops was impractical. If they were expected to become economically self-sufficient in their new homes, they would require continuous assistance, advice, and encouragement to develop their land for their own benefits, and contribute to the Protectorate's economy in the future. It was clear that the complexity of the problems involved in the movement of the settlers to a different geographical environment were not fully appreciated by the Colony administration and the greater financial responsibility it had to provide for the Gilbertese over a lengthy period of time than originally planned, was unavoidable. However, they should have not been convinced that life in the Protectorate was easy, but instead, should have been informed that their relocation to the Protectorate would involve intensive physical labor beyond levels that they have ever worked before in their lives. This should have begun in the Colony where they were made aware of the Protectorate, its geographical isolation and its climatic, environmental, and cultural conditions, prior to their departure, so that they could prepare themselves physically, psychologically, and materially for their new home.

### 4.4. Migrants and Recipients

Although the BSIP administration had a positive outlook on the Gilbertese resettlement to Solomon Islands, the indigenous population was not made aware of the arrival of the Gilbertese settlers which resulted in some social tensions and marginalization between the local population and the settlers. Up to this day, most of the indigenous Solomon Islands people still do not know how and why the Gilbertese ended up in Solomon Islands. The Colony and Protectorate administrations as facilitators of the Wagina Resettlement Scheme failed to establish any responsibilities to the Gilbertese and the indigenous Solomon Islands population, which should have been the basis of the resettlement schemes. The Gilbertese were unaware that their relocation to the Protectorate was to relieve the Colony government from further economic responsibilities, while being expected to contribute economically towards the Protectorate's economy.

The motives behind the resettlement was based on the notion that the Colony had little to offer in the future and on an unsighted belief that life in the Protectorate was easy and comfortable. While the desire for the Gilbertese resettlement could also be of assistance in the development of future successful resettlement schemes, the Gilbertese were denied a clear knowledge of what their relocation to the Protectorate would mean for them, and for the receiving country. The indigenous population of the Protectorate should also be informed about the scope and aim of the Gilbertese resettlement to the Protectorate. It would have been significant to obtain the support of the indigenous population right from the beginning of the scheme that the influx of the Gilbertese into Solomon Islands would by no means threaten their future. However, the Colony and Protectorate administrations failed to ensure that these obligations were fulfilled prior to the relocation, and, as a result, created hostility between the indigenous communities and the Gilbertese over the occupation of Wagina and access to education and employment opportunities in the Solomon Islands.

### 4.5. Land

The allocation of land and titles was also a major problem experienced by the Gilbertese. Prior to their relocation, the Gilbertese were advised by the GEIC administration to revert all the lands that they owned in Phoenix Islands in exchange of the land which would be allocated to them when they arrived on Wagina. The land assumed for settlement was inadequate for Gilbertese on Wagina and additional land had to be acquired from the offshore islands for coconut planting. However, the ownership of land allocated to the people on Wagina remains contested to this day. Their rights to the allocated land plots and ownership of Wagina are not legally recognized under the Solomon Islands Law up to this day, despite being acknowledged as occupants of the island. With no absolute rights over the land that they occupy and ownership of Wagina, the Gilbertese continue to feel insecure about their future and

the future of their children. Some of the village leaders have pursued the documented evidence of the Gilbertese ownership of Wagina, but so far, no documentation has been found. As a result, land issues continue to persist to this day with the proposed bauxite prospecting on the island, which has caused tensions between the Gilbertese, the government, and the prospecting company. The Gilbertese on Wagina, whose rights were never formalized when they arrived in the Solomon Islands, fear further displacement should any development activities proceed on the island.

*4.6. Identity*

The Gilbertese on Wagina have been continuously constructing and reconstructing identities according to their home of origin, ethnicity and citizenship. The people identify themselves as Gilbertese based on their home of origin and I-Kiribati based on their ethnicity, and as Solomon Islanders based on their citizenships. The Gilbertese strongly identify as I-Kiribati people because of the language they speak and the cultural practices and way of life they continue to observe in the Solomon Islands. Despite being identified as I-Kiribati in the Solomon Islands, this self-acclaimed identity is often perceived differently by I-Kiribati people from Kiribati, who consider them as Solomon Islanders. Whereas in the Solomon Islands, the local population still identify the Gilbertese people as 'Gilbertese' to identify them as people from the Gilbert Islands, or as *neiko. Neiko* in the Gilbertese language is used to refer to a woman or female, and therefore is used in reference to any Gilbertese. Multiple identities have been continuously constructed through intermarriages with indigenous Solomon Islanders, Asians, Europeans, and other ethnic groups. However, the older generation feared that the Gilbertese identity would one day vanish as more Gilbertese people intermarry with other ethnic groups especially indigenous Solomon Islanders as a mechanism to access to land, for social integration and permanency in the Solomon Islands.

**5. Implications of Displacement for the Pacific Islands**

While voluntary migration is most effective when decisions are made at the individual and household levels, forced migration leads to displacement (Mortreux and Barnett 2009; Brown 2012; Campbell 2014; McAdam and Ferris 2015; Ferris et al. 2011; Hugo 2012). The Gilbertese resettlement provides an example of migration as an adaptation strategy and migration as displacement. The relocation to the Solomon Islands was intended for the Gilbertese to escape the detrimental effects of droughts in their islands and improve their livelihoods in their new home. However, this example of adaptation was unsuccessful due to the lack of adequate planning and preparation by the GEIC and BSIP administrations that failed to ensure a smooth transition of the Gilbertese into their new home given the environmental and cultural differences of Wagina and Phoenix Islands. It was also evident that the relocation was orchestrated primarily to benefit both administrations economically and in demographic terms. While the BSIP administration was generous with its intake of the Gilbertese population, the GEIC administration saw it as a remarkable offer that could address the overpopulation issues in the Colony and free itself from further economic expenses of managing Phoenix Islands. On the other hand, the Protectorate's copra plantations had suffered labor shortages after the war. It was anticipated that the Gilbertese resettlement would not only benefit the Solomon Islands through additional population, but that they would also supply the required labor for the plantations, and contribute towards the Protectorate's economy in exchange for permanent settlement.

The Gilbertese settlers refused relocation to the Solomon Islands when they were informed by the GEIC administration. According to the Gilbertese on Wagina, many of them refused to leave their home islands, but believed that they were forced to be relocated to escape the aftermath impacts of Britain's nuclear activities on Christmas Island. According to an old man in his seventies who was part of the relocation to Wagina Island as a young man, '*my family refused to leave Orona because of the graves of our loved ones that would be left behind on the island*'. An old woman in her late sixties expressed her resentment towards the GEIC administration by articulating that due to '*the crowns of the coconuts trees fell after that thing [bomb] fell on Christmas Island. Yes, it was their doing*', she said, '*that's why we moved*

*here. They lied to us that our land is dead . . . we saw that the coconut trees were on fire because of that thing [bomb] that fell, it's like fire . . . so when the crown of the coconut trees fell, the place became dead'*. Although, the colonial records have been silent about these activities, there was evidence that Britain launched several hydrogen bombs between 1957 and 1959 on Christmas Island (Firth 1986).

The Gilbertese resettlement to Wagina was only made possible because the Colony had political, economic, and demographic ambitions in the schemes, without taking into account the impacts of relocation on the Gilbertese people as well as the Solomon Islands population. Today, the Gilbertese on Wagina have lived in Solomon Islands for more than 50 years, but they continue to experience cultural discrimination, land issues, and a lack of employment and education opportunities. Many have sought intermarriages to facilitate the integration of Gilbertese into the larger Solomon Islands society. However, many Gilbertese continue to see themselves as minorities, migrants and second class citizens in the Solomon Islands.

With the increase of climate change impacts, particularly sea level rise, migration, whether voluntary or forced, may be inevitable for many coastal communities, especially low lying atolls in the Pacific, given their low adaptive capacity and the lack of land to move further inland or uphill (Mortreux and Barnett 2009; Gemenne 2010). Although much has been written and discussed about migration as an adaptation strategy and migration as displacement by many scholars, very little has been documented about these forms of movements from the Pacific Islanders' point of view. The forced relocation of the Gilbertese to the Solomon Islands was described by the people on Wagina as being uprooted from their land by the GEIC administration and forced to relocate to Solomon Islands. They were very sad to leave behind their homes, the lands that they have worked hard to develop, and the graves of their loved ones buried on the islands. As one of the old men from Arariki village said; *'our hearts were very heavy when we sailed away from the islands. We could see our hard work and the graves of our loved ones left behind'*.

This grief reflects Pacific Islands' anxieties of being forced to relocate from their home islands if sea level rise continues to increase. Land plays an important cultural and spiritual role for many Pacific Islands. It is important because it is indicative of prestige, wealth, heritage, security, and identity; it represents people's birthplace and place of origin (Crocombe 1987; Bonnemaison 1985; Campbell 2014; Tabe 2011). The Gilbertese on Wagina expressed a sense of displacement because they felt that they were forcefully uprooted from their land and denied agency to decide whether they want to be relocated or remain in their home islands. This sense of being displaced is expressed by Pacific people through fears of being uprooted from their islands. To be 'uprooted' in the Gilbertese language symbolizes detachment from the land in which roots are entrenched. To be uprooted from the land signifies the loss of ties to the ancestral lands and history, cultural values and identity, and the sense of belonging. This means that the forced migration of Pacific Islands as a result of climate change will lead to displacement, even if they were to re-establish themselves in a new place or country. For example, Rudiak-Gould (2013, p. 157) observation of the Marshall Islands highlights that *'migration is not a solution to climate change, but rather a genocide for its people, because there will be no more Marshallese people, no language, and no culture'*. This was accentuated recently in a news statement by the President of the Marshall Islands, Hilda Heine, where she reiterates that *'her government won't consider relocating her citizens in responses to climate change'* (RNZ 2018).

However, migration may be inevitable for Pacific Islands if sea level rise continues to increase. If so, planned relocation will be essential because most of these movements will take place internally within the state's territory. Although it is the state's responsibility to facilitate all planned relocations within its national borders, it should only be carried out if all other alternatives and solutions have been exhausted and relocation is decided by the communities as the preferred form of adaptation to sea level rise. Planned relocation should enable the rebuilding, integration, and sustainability of relocated communities in the new destinations and should only take place with informed consent of the vulnerable communities and the people (UNHCR 2014). For instance, Tuvalu has emphasized on the fact that 'there is no Plan B', which is to be relocated. The country is working on building

climate resilience and to enhance the adaptive capacity of its people to adapt better to the impacts of climate change. If they have to migrate, it will be the very last response after every available option to remain on the island has been explored and exhausted (Mortreux and Barnett 2009; Campbell 2014; Ransan-Cooper et al. 2015).

However, it has been argued that migration as an adaptation strategy to climate change should not deter the focus of developed countries from cutting down their carbon emissions, because for Pacific Islands, forced migration will lead to displacement. Instead developed countries should take responsibility in mitigating climate change and assist the Pacific Islands in building innovative adaptation strategies so that the islands remain habitable for many years to come. It is also important that Pacific Islands' narratives on forced relocation and displacement of their people from their land are acknowledged and privileged in climate change discourses and policy making. This is not to completely reject migration as an adaptation strategy, but rather to recognize Pacific Islands' voices and perceptions on migration as displacement for them. This is also to help inform decisions on how forced migration of vulnerable communities in the Pacific can be carried out effectively and in line with best practices for future relocations that have been developed by the United Nations of High Commissioner for Refugees (UNHCR 2014) to reduce risks associate with climate change and ensure the continuity of communities in the new destinations.

## 6. Conclusions

Campbell (2014) clearly argues that when discussing migration, it is important to differentiate between 'migration as adaptation' and 'migration as displacement'. The Gilbertese resettlement to Solomon Islands provides a relevant example of forced relocation and displacement of Pacific Island people during the colonial period. It also illustrates an example of the relocation of people as an adaptation measure, but also as one that led to the displacement of the Gilbertese people. The resettlement offers some lessons that can inform decisions and policies on climate change migration and relocation, especially of vulnerable Pacific Islands that may be subject to relocation in the future as a result of sea level rise. The Gilbertese resettlement provides lessons that implicates the need for planned relocations to involve proper planning and preparation with informed consent from the communities and people. People who may be subject to forced relocation in the future as a result of climate change must be given the agency to choose whether they want to be relocated or to remain behind to minimize potential challenges likely to be encountered in the new destinations and a sense of displacement.

The environmental and cultural differences of the islands need to be taken into account during the planning and preparation of any community relocation to minimize the challenges that are likely to be encountered. Consultation with the host communities prior to any relocation (unless the relocation site is within the community's land boundaries) is significant so that they can provide the relevant support needed in the integration of the communities in their new homes and to prevent any form of hostility and marginalization. Land is evidently a major issue in migration processes and discourses, especially in the Pacific Islands. Land should be acquired legally with rights and allocated to the relocated communities so that they can re-establish themselves and continue to exist as a community. Rights and access to fishing grounds should also be allocated to relocated communities that are dependent on marine resources to provide them means for subsistence and income-generating activities. Identity is also very important to many Pacific Islands and it is one of the reasons why many do not want to migrate. However, it is important that relocated communities continue to thrive in the new destinations so that they can maintain their cultures, languages, and maintain their identity.

While a lot has been documented and discussed about migration as an adaptation strategy and migration as displacement, very little has been written from the Pacific Islands standpoint. Many Pacific Island countries view forced migration or relocation as a form of displacement. This is not to dismiss migration as an adaptation to climate change, but rather to understand why many Pacific Island countries resist the idea of being forced to leave their home islands and relocate elsewhere due to

climate change. This resistance embodies the fear of losing their genealogies, cultures, and identity imbedded in the land. Land is culturally and spiritually important for Pacific Islanders and is greatly valued, and to be forcefully uprooted from the land leads to displacement. The resistance towards forced relocation also challenges developed countries must reduce their global carbon emissions so that Small Island Developing States (SIDS) like the Pacific Islands should not become displaced from their home islands as result of sea level rise. It also accentuates Pacific Islands' voices and position in global discourse as small island states for developed countries to take responsibility in curbing their carbon emissions. If migration was to take place, it must be carefully planned and conducted properly with consent of the communities and people. The relocation must be planned and facilitated in a way that enables the continuity and sustainability of the community in the new destinations. However, the recognition and support of Pacific voices and perceptions in discourse on climate change migration and displacement is crucial at regional and global forums, particularly to inform decisions and policies for effective planning of future climate change migrations in the Pacific region.

**Funding:** This paper received no external funding.

**Acknowledgments:** This paper is part of a Ph.D. thesis at the University of Bergen, Norway. Many thanks to Akosita Rokomate-Nakoro for the constructive feedback on the paper, and to John R Lee for the continuous encouragement.

**Conflicts of Interest:** The author declares no conflict of interest.

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
