# Peer review of "Climate Change Migration and Displacement: Learning from Past Relocations in the Pacific"

_socsci, doi:10.3390/socsci8070218_

Round 1

Reviewer 1 Report

Thank you for this interesting article.  The history of the relocation of i-Kiribati people to the Phoenix and then Solomon Islands has been very much overlooked with only a few academic articles on the topic.  Of these, quite often the issue of the Gilbertese is not the main focus of the articles (for example it might be minority populations in Pacific Island countries).  So on its own the merits the story needs to be told.  It is even better that the story is from the resettled people themselves and is presented by one of their descendants.  The case becomes even more important when we consider its relevance to possible climate change responses in the Pacific Islands.

I have a few of concerns that would be useful to address.  The first of these is the use of the term displacement.  I think the paper needs to explain what is the difference between displacement and migration or forced migration.  For example, on line 20 it is stated ‘migration can lead to the displacement’.  On line 268 it is stated ‘forced migration creates displacement’.  My thought was, isn’t forced migration a form of displacement?  On lines 381-382 there is a statement that ‘While a lot of emphasis has been placed on migration as an adaptation strategy, it is displacement for many Pacific Islands’. Perhaps the terms displacement and (forced) migration need to be explained a little more fully and the distinctions between them clarified.  A few sentences would do the job perhaps in the introduction.

My second issue relates to voices from the community of resettled people.  I understand that you are speaking for your elders, but it would be really nice if you could include some direct quotes from your ethnological research.  As you say in the conclusion, ‘It is also essentially important that emphasis is given to Pacific Islands narratives on migration’.  I couldn’t agree more.  This paper is itself a Pacific Island narrative and it is very important that your voice is heard.  But if possible some quotes (just a couple or three perhaps)from the people in Wagina would help stress the emotional and spiritual impacts that the relocation has had on them.  I think many readers struggle to understand the essential nature of the human/land bond in PICs and their voices may help improve this understanding.  Perhaps these quotes could go in sections 4 and/or 4.1.

Third, it would be perhaps useful to have a paragraph that outlines the physical differences between Wagina and the atolls from which the people were relocated.

There are quite a few grammatical and other mistakes that I have listed in an attached document.  It also lists some queries about thinks that may need to be clarified or some suggestions that you may wish to consider.

Best wishes with your work.

Author Response

Dear Reviewer 1- I have revised my paper and incorporated the recommended corrections from you. Please see my revised paper and let me know whether or not I have addressed your comments accordingly.

Reviewer 2 Report

The paper contains a lot of interesting information and some good analysis. However, in its current form I do not recommend for publication as there are several significant issues which need to be addressed.

1.      The paper would benefit through early and consistent definitions of terms such as migration and displacement. One way of overcoming this would to use the emerging definitions used within UNFCCC processes.

2.      It is unclear for the reader where the results come from. The method section briefly talks about archives, interviews and a survey but should be explained – I don’t doubt that the combination of documents and fieldwork is appropriate, but this needs to be briefly explained and justified. The results do not stipulate any relationship with fieldwork so it reads like a literature review.

3.      It would be useful to consider some best practices of relocation in the region or elsewhere. Perhaps start by considering UNHCR’s work on this with Brookings.

4.      The language needs work. Paragraphs are generally too short and do not contain fully developed ideas. It is also necessary to do a thorough proof-read; there are even typos in the abstract.

Other comments

9-10 Migration as a form of displacement? Is this the wrong way around? Also quite a lot has been written on this theme so it is strange for you to claim it is so novel.

16-17 This is confusing as in my understanding, forced migration = displacement

25-29 Confusing sentences right at the start of the paper.

44 “America” = “USA”?

49-50 and 58-59 both reference natural hazards so the paragraph is repetitive. This occurs several times in the paper.

69 “National” contains an upper case “N”. This also occurs in other places.

72 “stormed” = “impacted”.

97-99 75 million is not a very useful statistic as the vast majority refers to Asia, as opposed to Pacific Islands.

114-125 Very confusing use of terms.

165 how many interviews. What sort of interviews?

174-176 why is this included in the methods?

267-271 these distinctions should be in the introduction as central to your argument.

345 this section is powerful, but I would avoid the term “genocide” unless you explain that is someone else’s opinion.

Author Response

Dear Reviewer 2 - I have tried to address all your comments in my revised paper, but do please let me know if I have not done so I can revise the paper again.

Reviewer 3 Report

Wagina Resettlement

This is a potentially useful account of a largely forgotten population movement in the Pacific, and as the author quite rightly says, there are lessons to be learned here that have potential repercussions elsewhere. But very considerable rewriting is needed first. There are two basic problems. Firstly, much of what is said here has been said many times before and is scarcely new, and, secondly, the actual account of the Gilbertese relocation to Wagina (which is new) is covered in just two pages– and that too is very vague in places.  Ideally then this section should be significantly expanded.

The background to issues of resettlement (pp. 1-3)  is quite well-known but the coverage is somewhat idiosyncratic, and needs to differentiate more clearly between pre-colonial resettlement and post-colonial resettlement, and to reflect on whether changes resulting from mining (e.g. the unusual case of Banaba) do or do not have implications for resettlement following climate change (and whether that climate change refers to hazardous events – e.g. tsunamis or cyclones – or to long-term sea level rise (SLR) etc).

There are as yet no examples of resettlement of people in the face of long term SLR (but the unpopulated islands off eastern Choiseul are suggestive of change, or at Taro, at the other end of Choiseul – that should be mentioned – odd that they are not when so close to Wagina, and quite well documented). Some people have certainly moved   (away or inland) after cyclone events – and, again there are several studies of this in Fiji, which, again, are curiously absent from this study, yet very relevant to climate change. It is however a total exaggeration to talk (lines 70-71) about the ‘death and displacement of many people .. and many examples across the Pacific’. If there are they need to be specified here. ‘Forced’ relocation is rare – and there are few if any examples (and then mainly as a result of moving people for mining, such as at Banaba, or the well-known movement of people from Bikini atoll). As the author points out (p. 2) – even the victims of volcanic eruptions generally returned home when it was safe to do so.

Frustratingly the actual account of relocation is so brief as to be of little value. It mostly discusses whether the Gilbertese wished to go or not, and labels it a case of forced migration. But were they forced? It was my understanding that the first move from the main Gilberts chain to the Phoenix Islands was largely voluntary (in moving from somewhat drought prone atolls with a high population density) and then, when the Phoenix scheme failed (because of drought and distance), people were anxious to be resettled elsewhere, but could have chosen to return to the Gilberts.

What is crucial here is an elaboration of the paragraph starting ‘Unfortunately..’. What were the problematic physical, political and socio-economic factors, and were they experienced by all households? How many households moved? Did leaders go with them? Were they consulted/involved? What islands were they from originally? What sort of land, with what potential were they offered/ given? How was land distribution done? (Problems are hinted at – lines 302-3). Were markets accessible? Was housing provided, on what terms, and other infrastructure, such as water etc?  Were they supported in fishing activities .. etc etc. Lack of advice and assistance was certainly not unusual – but what was particularly problematic? Was it a general fault of colonial planning? And what experience did colonial authorities build on? Did men have to work on plantations – as hinted at? What lessons learned in the Phoenix resettlement did the settlers bring with them so that they could have more effectively exerted their own agency?  Did all household conceive the move equally negatively or were some happy and successful? etc etc   Nothing of this is in the paper yet that is what is clearly needed so that lessons could be learned from this case.

The author states that (s)he interviewed mainly elders – the group most likely to be nostalgic for a past. Were younger people more positive? Is it possible for older people anywhere to be content with long-distance migration moves? (This I however curious since the move took place seemingly almost (?) fifty years before the fieldwork – when these elders would have been children).

On their own island, why were conflicts with the SI populations so problematic? That was more likely to be true in Gizo and the Shortlands.

The conclusion is weak since there are only generalities again, frustratingly repeating the same vague themes about isolation and the loss of land and culture. This is more likely to occur again in the future, for Kiribati and elsewhere, and we need more guidance from the past in what can be learned to avoid or minimize this. What would the author suggest that Kiribati, for example, now do to avoid the problems of the past?

Other points:

p. 1 Reference to Hau’ofa has become something of a politically correct cliché. He offered little more than rhetoric and other scholars (e.g Paul D’Arcy) have challenged the notion of the ever mobile and connected islanders. Indeed where D’Arcy mainly worked – Micronesia, and therefore Kiribati – all the evidence suggests that people moved only out of necessity (e.g. after hazards) and it was not always benevolent (The prehistory of Kiribati is one of warriors). Indeed if there is ‘forced migration’ these early times are when it occurred.  

Lines 34-37  Rather than administrative empires it was the impact of capitalism that resulted in migration – whaling, plantations etc – but that needs differentiating (and is all doubtfully relevant to this paper).  

p. 3  need to update IPCC data from the earlier study

p. 101 Why does remoteness make countries vulnerable to climate change?

lines 109-112  All atolls are low lying (at least in Kiribati) There seems to be some confusion between atolls and islands?

lines 119-121  This tautologous sentence makes little sense

p. 6  line 196 Implies that it is Phoenix Islands that is Kiribati – not so.

p. 7  line 240  Curious that isolation was a problem after the Phoenix Islands? In what way? Not so far from Titiana? (which should be on Fig 1).

p. 10  The paragraph on Tuvalu is totally irrelevant. The next paragraph also loses focus when it should be focusing on what might be learned from the Wagina experience.

There are far too many typos and grammatical errors to specify.. but, as one tiny example,  it would be good to spell ‘indigenous’ correctly (line 289). Some references (e.g. Weber 2015) are not in the Bibliography.

Two studies that would be useful are:

Cochrane, G. (1970). The Administration of Wagina Resettlement Scheme. Human Organization, 29(2), 123-132.

Knudson K (1977). Sydney Island, Titiana, and Kamaleai: Southern Gilbertese in the Phoenix and Solomon Islands. In Lieber M, ed, Exiles and migrants in Oceania. Honolulu: University of Hawaii Press, pp. 195–242.

Author Response

Dear Reviewer 3 - I have tried to address all your comments in my revised paper but please let me know if I had not done so.

Round 2

Reviewer 3 Report

Wagina Rsettlement

This is now much improved and has been extensively rewritten, especially with the necessary detail on the Wagina resettlement. It is incidentally usually courteous and necessary to indicate what responses have been made. With a few minor changes I think it is now publishable. The opening is more solid and the conclusions are more directed to resettlement issues rather than irrelevant assertions from elsewhere. Good that Fiji is now included – but this probably that should be returned to in the Conclusion – is there any sense in which the Fijian resettlement has benefited from knowledge of the kinds of things that went wrong at Wagina? Is resettlement more successful now that half a century ago? Or are Fijian circumstance different?  At last we are now getting more of the story of resettlement (before the last original resettlers die off!). I think it can be published once the flowing issues have also been addressed.

Fig 1  Titiana should be on the map

Still no reference to Taro and the smaller Choiseul islands that disappeared.

Line 210  providing

Line 212  Christmas Island (not Islands) is irrelevant to the paper.   

Line 240  mature (not sure what this means though)

Line 364  Nikumaroro

Line 387  intensively hard?

Line 394  Best to say Melanesia rather than fizzy – but worth nothing that western Solomon Islanders are some of the blackest in the world

Line 464 verb missing

Lines 489-91   Christmas Island again -  resettlement to CI only occurred much later. Given its even greater remoteness (hence the nuclear testing) would it have made a site? The text implies that there was resettlement from CI, but obviously not. Better to leave out this speculation.

Line 508  Already said on line 378.

Line 520 isolated

Line 537 Its

Line 578 Sentence is ungrammatical.

Line 679  response

Line 688  Sentence is ungrammatical.

Line 693  Ransan-Cooper

Author Response

Please see attached file for revisions addressed in the text based on all the comments provided. 

Reviewer 3 Comments and Revisions

The opening   is more solid and the conclusions are more directed to resettlement issues   rather than irrelevant assertions from elsewhere. Good that Fiji is now   included – but this probably that should be returned to in the Conclusion –   is there any sense in which the Fijian resettlement has benefited from   knowledge of the kinds of things that went wrong at Wagina? Is resettlement   more successful now that half a century ago? Or are Fijian circumstance   different?

The   Fijian circumstance is very different from the Wagina relocation but they   both provide lessons that can inform future relocations – relocation within a   community’s land boundary in the case of Vunidogoloa and relocation beyond   national jurisdictions in the case of the Gilbertese people.

Fig 1  Titiana should be on the map

This   has been included on the map

Still no reference to Taro and the smaller Choiseul   islands that disappeared.

Reference   to the relocation of Taro township has been made in the text.

No   reference was made to the disappearing islands in Choiseul because at this   stage they are not relevant to the focus of the article.

Line 210  providing

This   has been addressed in the text

Line 212 Christmas Island (not Islands) is   irrelevant to the paper.   

This   section has been removed from the text

Line 240  mature (not sure what this means though)

This   has been amended in the text

Line 364  Nikumaroro

This   has been amended in the text

Line 387  intensively hard?

This   has been reworded in the text

Line 394  Best to say Melanesia rather than fizzy   – but worth nothing that western Solomon Islanders are some of the blackest   in the world

This   has been reworded

Line 464 verb missing

Changes   were made to the sentences along this line.

Lines 489-91   Christmas Island again -    resettlement to CI only occurred much later. Given its even greater   remoteness (hence the nuclear testing) would it have made a site? The text   implies that there was resettlement from CI, but obviously not. Better   to leave out this speculation.

These   lines have bene removed from the text.

Line 508  Already said on line 378.

This   has been removed

Line 520 isolated

This   has been included in the text

Line 537 Its

This   has been included in the text

Line 578 Sentence is ungrammatical.

This   has been revised in the text

Line 679  response

This   has been included in the text

Line 688  Sentence is ungrammatical.

This   has been revised in the text

Line 693  Ransan-Cooper

This   has been corrected in the text